# Spatial variation in socio-economic vulnerability to Influenza-like Infection for the US population

Shrabani S. Tripathy[1], Joseph V. Puthussery[1,2,3], Taveen S. Kapoor[1], John R. Cirrito[3], Rajan K. Chakrabarty[1]*

1 Center for Aerosol Science and Engineering, Department of Energy, Environmental and Chemical Engineering, Washington University, St. Louis, Missouri, United States of America, 2 Department of Civil Engineering, Indian Institute of Technology Delhi, Hauz Khas, New Delhi, India, 3 Department of Neurology, Hope Center for Neurological Disease, Knight Alzheimer's Disease Research Center, Washington University, St. Louis, Missouri, United States of America

* chakrabarty@wustl.edu

## Abstract

This study aims to quantify environmental health impacts and assess risk by understanding the disproportionate burden of infectious diseases, specifically Influenza-like Illness (ILI), across regions with varying socio-economic characteristics. We introduce a novel vulnerability-based approach to better understand the complex relationship between socio-economic factors and ILI burden. We develop a machine-learning-driven framework to assess and map state-level socio-economic vulnerability to ILI in the United States. A vulnerability index was created by integrating 39 diverse socio-economic and health indicators from the latest CENSUS. A Random Forest Regression model then weighed these indicators to quantify each state's vulnerability for the ILI values in 2022. To assess multicollinearity, Variance Inflation Factor (VIF) was calculated, and parameters were filtered to reduce the VIF. Key determinants of vulnerability include migration patterns, insurance coverage, and proportions of female and elderly populations. The resulting state-level vulnerability map reveals significant regional disparities. District of Columbia was identified as the most vulnerable state, followed by Massachusetts, Hawaii, New Mexico, and Rhode Island, all with normalized vulnerability indices exceeding 0.35. Our findings highlight significant regional variations in ILI vulnerability, emphasizing the need for targeted public health interventions tailored to state-specific socio-economic conditions. This scalable and adaptable methodology extends beyond influenza, offering a valuable approach for assessing vulnerability to a wide range of infectious diseases, strengthening epidemic preparedness and response.

**Data availability statement:** All publicly available data used in this manuscript are appropriately cited in the main text and are as follows: CENSUS data: https://www.census.gov/ CDC data: https://www.cdc.gov/fluview/. The data is also available at: https://data.mendeley.com/datasets/v4b2w26jbc/1.

**Funding:** This work was supported by the FluLab (GR0031819 to RKC and JRC), Washington University's Transdisciplinary Institute in Applied Data Sciences (TRIADS) Seed Grant program (JVP, JRC and RKC). and the WashU-IITB Joint Research and Education Initiative (SST and JVP). The funders had no role in the study design, data collection and analysis, the decision to publish, or the preparation of the manuscript.

**Competing interests:** The authors have declared that no competing interests exist.

## Author summary

Socio-economic conditions significantly influence a location's susceptibility to Influenza-like Illness (ILI). Identifying infection hotspots based on these factors and quantifying relative susceptibility is crucial for policymakers. A vulnerability index is an effective tool for this purpose. Our study employs a Random Forest-based technique to not only provide relative vulnerability values but also to identify the dominant socio-economic features contributing to high vulnerability in specific areas.

## 1. Introduction

Vulnerability assessment is a crucial tool for understanding how populations experience and respond to external threats, ranging from natural disasters [1,2] to economic crises [3,4] to long-term impacts of climate change [5,6]. Defined as the degree to which a system or population is susceptible to harm and lacks resilience, vulnerability is shaped by a combination of socio-economic, demographic, and geographical factors [7,8]. The Intergovernmental Panel on Climate Change (IPCC) defines vulnerability as the degree to which a system or person is susceptible to harm or damage from climate change [9]. Vulnerability has three connotations: it refers to a consequence rather than a cause, implies an adverse consequence, and is a relative term that differentiates between socio-economic groups or regions, rather than an absolute measure of deprivation [10,11]. The construction of social vulnerability is shaped less by geographical location and more by inherent socio-economic structures [12–14]. Originally framed in the context of environmental hazards, this concept extends to public health, where vulnerability reflects a population's susceptibility to infectious disease burden and its ability to cope with health risks. Consequently, vulnerability assessments have been widely applied in hydroclimatic contexts such as floods [15,16], droughts [1,17], and sea-level rise [18]. However, their application in public health and infectious disease risk assessment remains underexplored. For example, some regions with high population density, limited healthcare access, or low income may be more vulnerable to health crises than others with lower population density or higher average income. Vulnerability assessment for health risk then allows the relative susceptibility of these different regions to be quantitatively understood and compared. Given the growing recognition of socio-economic determinants in shaping health outcomes [19–21], a structured vulnerability-based framework for disease analysis can provide a more holistic understanding of susceptibility beyond traditional epidemiological approaches.

Health risk is not solely determined by pathogen exposure but is also influenced by a range of socio-economic factors that govern individuals' ability to prevent, respond to, and recover from infections. Factors such as income disparity, healthcare access, housing conditions, education, and employment status play a critical role in shaping regional variations in disease transmission and adverse health outcomes

[22–24]. While past studies have examined correlations between individual socio-economic variables and disease burden [25,26], few have developed a quantitative framework that integrates these factors into a panoramic vulnerability model. This study seeks to address this gap by developing a vulnerability-based assessment of Influenza-like Illness (ILI) in the United States, leveraging a machine learning approach to quantify and map state-level susceptibility.

ILI represents a significant public health challenge due to its high transmissibility and seasonal recurrence. Caused primarily by influenza viruses (types A and B) and respiratory syncytial virus (RSV), ILI imposes substantial morbidity and economic burden, particularly among socially and economically disadvantaged populations [27,28]. The Centers for Disease Control and Prevention (CDC) defines ILI as a fever of at least 38°C accompanied by a cough or sore throat within the past 10 days. The annual cost of influenza in the U.S. is estimated at $11.2 billion, including both direct medical expenses and productivity losses [29]. However, the ILI burden is not uniformly distributed across states but is shaped by systemic disparities in socio-economic conditions, healthcare infrastructure, and population density. Individuals residing in high-poverty, overcrowded, or medically underserved areas face a disproportionately higher risk of infection, hospitalization, and adverse outcomes [30–32]. Despite this well-documented association, there is no standardized framework that quantifies ILI vulnerability at a state level, incorporating both socio-economic and healthcare accessibility factors.

In contrast to traditional analyses of disease vulnerability that focus on individuals or specific groups, this research adopts a regional perspective. In this study, we aim to understand how a state's overall vulnerability is shaped by aggregated socioeconomic indicators. This approach considers the interconnectedness of factors within a defined geographic or administrative area. By using indicators such as state level poverty rates, average educational attainment, and access to healthcare facilities per capita, we characterized the overall vulnerability of distinct states. This regional analysis facilitates spatial comparisons, identifies vulnerability hotspots, and provides valuable insights for policymakers and planners seeking to develop targeted strategies at a broader scale.

The CDC provides a commonly used Social Vulnerability Index (SVI) for assessing risk to natural disasters [33]. It quantifies the degree to which a social condition may affect a region's ability to prevent losses in a disaster. Based on 16 social indicators (4 themes), CDC ranks each census tract and aggregates the ranks to determine the final ranking of each location. Though this index provides a basic understanding of the social vulnerability of a region, it has a major drawback. The index is generalized and treats responses to all natural disasters alike. Furthermore, it considers the individual values of each indicator to rank the vulnerability, assuming a uniform weight. However, in the case of a real disaster, different parameters play different roles based on the type of disaster. For example, during a coastal hurricane, home ownership and the age of housing stock become primary drivers of total loss. However, in the case of a non-structural event like a pandemic, factors like access to healthcare and the number of people with health insurance may drive the total loss. To overcome these drawbacks, here we propose a novel approach to use socio-economic indicators in a machine-learning-based approach to compute socio-economic vulnerability specific to ILI. Unlike the SVI, our machine learning algorithms can learn complex, non-linear relationships between socioeconomic factors and vulnerability, and can be trained on specific outcomes ILI rates.

Traditional approaches to disease vulnerability analysis have primarily focused on individuals or specific groups, identifying factors associated with increased infection risk [34–36]. These methods of identifying factors associated with increased infection rates have both advantages and limitations [25,26,31,37]. These methods often overlook the broader regional context. Most existing health risk studies treat the CDC's Social Vulnerability Index (SVI) as a standard baseline, often employing association-based analyses to understand the correlation between the SVI's parameters, or the overall SVI score itself, and various health hazards [38–41]. While these studies highlight the usage of machine learning approaches in establishing relationships between these socio-economic parameters and infection rates, they often rely on a single, general SVI score or the parameters used in SVI computation to represent vulnerability to any health crisis. In contrast, our approach employs a machine learning method to establish a unique, risk-specific relationship for influenza infection, considering a distinct set of socio-economic indicators. This research emphasizes the importance of an

indicator-based vulnerability assessment at the regional level, which captures the interconnectedness of socioeconomic factors within geographic or administrative boundaries. A regional perspective gives a more wide-ranging understanding of vulnerability, providing valuable insights for policymakers and planners. The rapidly expanding literature on social determinants of health in the U.S. and globally underscores the importance of addressing socio-economic disparities to reduce vulnerability and improve health outcomes [12,22,23,32,42].

This study introduces a novel machine-learning-driven vulnerability framework to systematically assess and map state-level socio-economic vulnerability to ILI in the United States. We use state-specific ILI data for the year 2022 (Fig 1), which corresponds to the period for which most of the socio-economic data is available. Unlike conventional epidemiological models, which primarily focus on disease incidence and risk factors, this approach integrates multiple socioeconomic and health indicators into a comprehensive vulnerability index. Using Random Forest Regression, we identify and weigh the most influential predictors of ILI susceptibility, allowing for the quantification of state-level vulnerability scores. The resulting vulnerability maps provide a spatial representation of ILI susceptibility, highlighting regional disparities and socio-economic drivers of disease burden. By adopting a vulnerability-centric perspective, this study offers valuable insights for targeted public health interventions, resource allocation, and epidemic preparedness beyond influenza.

## 2.  Results

The vulnerability of each state is quantified using selected socio-economic and health indicators. A Random Forest Regression (RFR) model assigns feature importance values to these indicators, which are then used as weights along

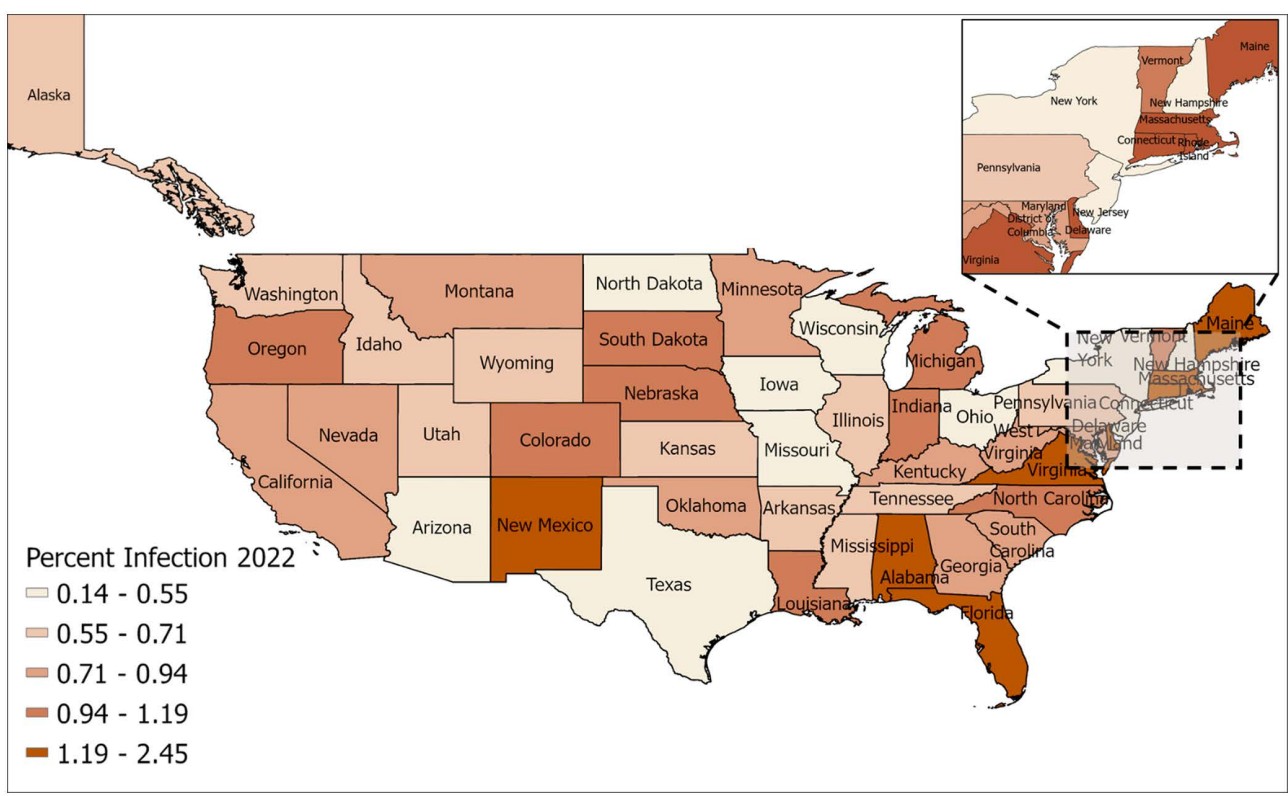

**Fig 1.  Influenza-like Illness (ILI) Peak Infections: State-level peak weekly ILI cases in 2022, expressed as a percentage of the total population.** The maps were generated using ArcGIS Pro using shape files from the U.S. Census Bureau (https://www.census.gov/geographies/mapping-files/time-series/geo/carto-boundary-file.html).

PLOS Computational Biology

with directionality derived using Partial Dependence Plots (PDPs), to compute each state's vulnerability score. Mapping these values spatially (Fig 2) facilitates understanding regional disparities, providing policymakers and stakeholders with a clear representation of vulnerability distribution to guide targeted interventions.

The normalized vulnerability index for U.S. states is categorized into five classes based on the quantiles: Very Low (0.0-0.13), Low (0.13-0.21), Medium (0.21-0.28), High (0.28-0.35), and Very High (0.35-1.0). This ensures that each category contains approximately one-fifth of the total states, or around 10 in this case. States with higher vulnerability, represented by darker blue shades, face greater socio-economic and health challenges, necessitating targeted public health interventions and resource allocation. The District of Columbia has emerged as the most vulnerable state to ILI. This spatial representation enables the identification of high-risk regions and supports data-driven decision-making for resilience planning.

To evaluate whether unmodeled spatial processes might influence our results, we computed Global Moran's I and Local Indicators of Spatial Association (LISA) using the centroids of the U.S. states. As presented in S1 Fig, the residuals exhibit a weak but statistically significant negative spatial autocorrelation (Moran's I = −0.0435, p = 0.0041), indicating spatial dispersion rather than clustering. This suggests that neighboring states tend to have dissimilar residual values, implying that the RFR model effectively captures most of the spatially structured variability in ILI vulnerability.

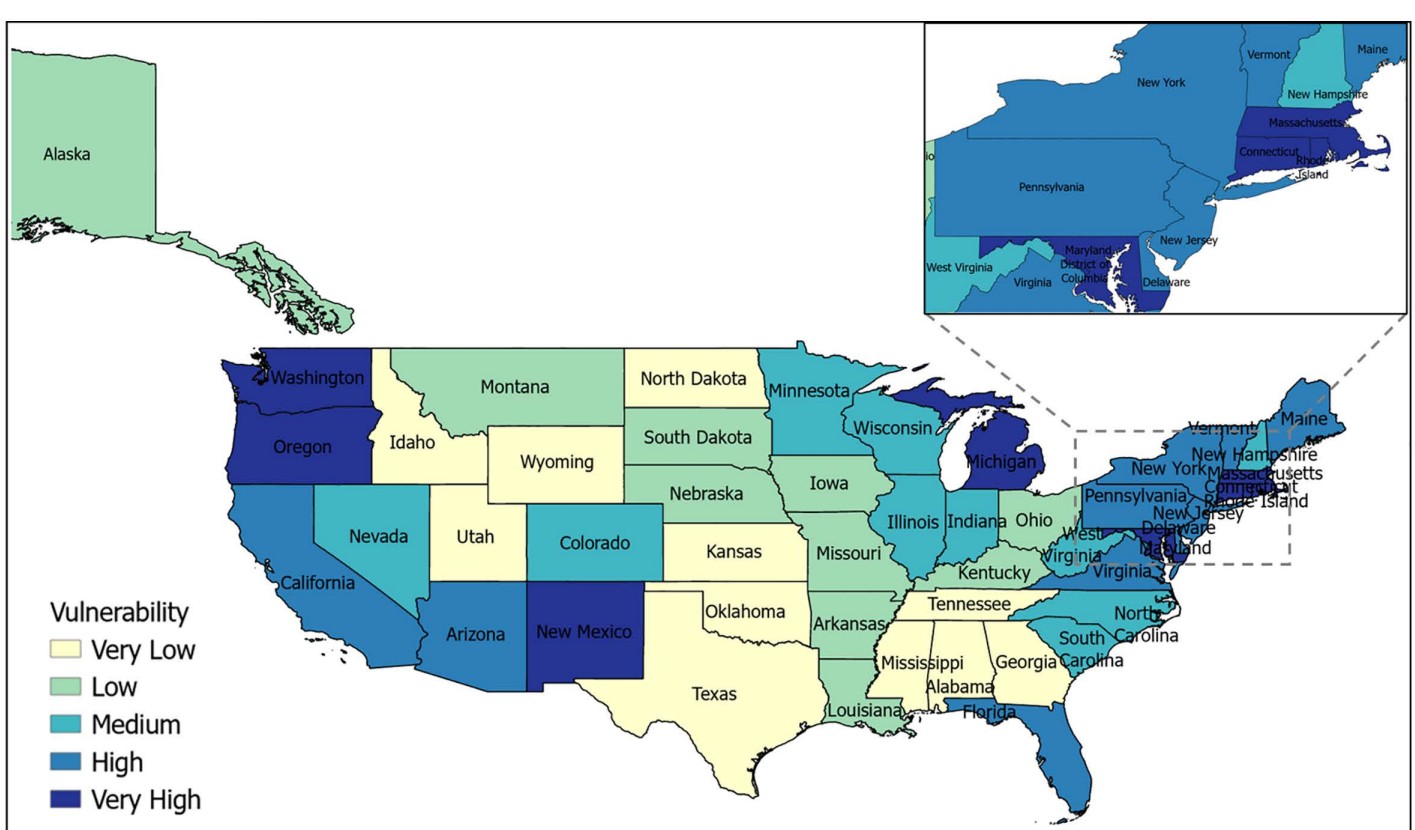

**Fig 2. U.S. ILI Vulnerability Map: State-wise vulnerability maps across the United States, highlighting regions with varying levels of socio-economic vulnerability to Influenza-like Illness (ILI).** States are classified as follows (normalized indices in parentheses): Very Low (0.0–0.13), Low (0.13-0.21), Medium (0.21-0.28), High (0.28-0.35), and Very High (0.35–1.0). The maps were generated in ArcGIS Pro using shape files from the U.S. Census Bureau (https://www.census.gov/geographies/mapping-files/time-series/geo/carto-boundary-file.html).

Based on the state-wise vulnerability values (Eq 3), we see that the District of Columbia, Massachusetts, Hawaii, New Mexico, Rhode Island, Connecticut, Maryland, Oregon, Washington, Michigan, and Arizona belong to the very high vulnerability category. To better understand the influence of variables, making each of these states highly vulnerable, we consider three distinct state types—an urban-dense state (District of Columbia), a mixed urban-rural state (Louisiana), and a predominantly rural state (West Virginia) (Fig 3). The plots highlight distinct socio-economic profiles contributing to the varying ILI susceptibility across diverse state types. Despite all three exhibiting high vulnerabilities to Influenza-like

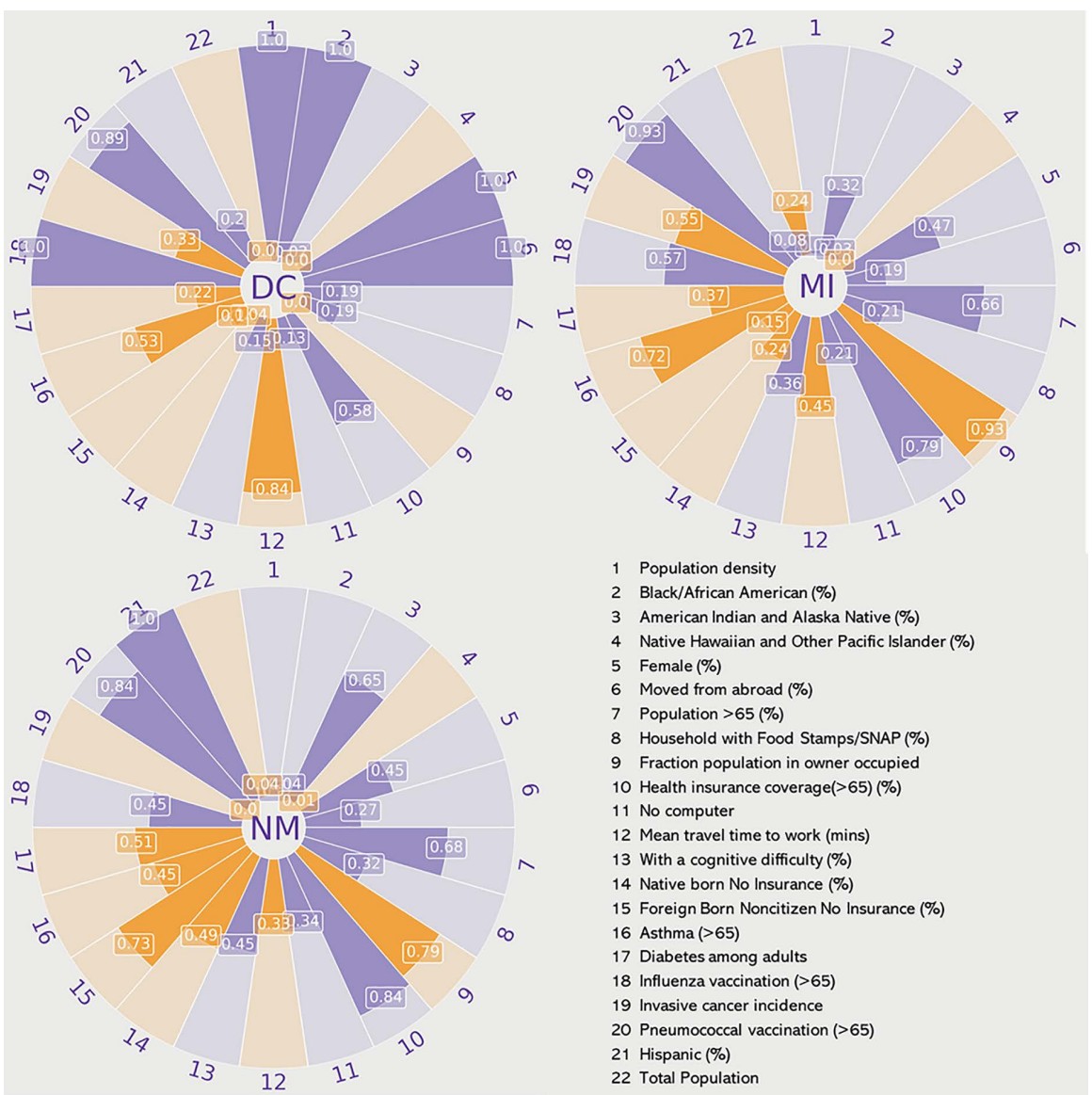

1  Population density
2  Black/African American (%)
3  American Indian and Alaska Native (%)
4  Native Hawaiian and Other Pacific Islander (%)
5  Female (%)
6  Moved from abroad (%)
7  Population >65 (%)
8  Household with Food Stamps/SNAP (%)
9  Fraction population in owner occupied
10 Health insurance coverage(>65) (%)
11 No computer
12 Mean travel time to work (mins)
13 With a cognitive difficulty (%)
14 Native born No Insurance (%)
15 Foreign Born Noncitizen No Insurance (%)
16 Asthma (>65)
17 Diabetes among adults
18 Influenza vaccination (>65)
19 Invasive cancer incidence
20 Pneumococcal vaccination (>65)
21 Hispanic (%)
22 Total Population

**Fig 3. Indicators for vulnerable states: This figure presents radar plots to compare the normalized values of the 22 indicators across three states very highly vulnerable to Influenza-like Illness (ILI): (a) District of Columbia (DC) (urban), (b) Michigan (MI) (mixed), and (c) New Mexico (NM) (rural).** The value on each section represents the indicator's normalized score (0 to 1). The Purple segments indicate factors that increase vulnerability (positive directionality), and orange segments indicate factors that decrease vulnerability (negative directionality).

Illness (ILI), the underlying causes differ based on state-specific policies, healthcare infrastructure, and socio-economic conditions.

Based on the vulnerability indicators, in the District of Columbia (Fig 3a), high population density, increased mobility, and a significant proportion of foreign-born noncitizens without insurance contribute to its susceptibility. Longer commute times and a diverse population further increase exposure risks. While the District of Columbia has relatively better health-care coverage and vaccination rates than other states (with a normalized value of 1), disparities in healthcare access, particularly for uninsured individuals, remain challenging. With a combination of urban and rural populations, Michigan (Fig 3b) faces high vulnerabilities because of a different set of factors. High older, female, and black/African American populations, and lower vaccination coverage, exacerbate vulnerability. Urban centers contribute to increased disease transmission, while rural areas struggle with limited healthcare access and the presence of a more susceptible category of population (older, female etc.), creating a dual challenge for public health management. Meanwhile, New Mexico (Fig 3c) demonstrates vulnerabilities linked to its aging, American Indian, and Hispanic population, high poverty rates, and limited access to mobile phones and computers. Suggesting that specific demographic groups, combined with limited access to digital health information, are key drivers of risk in the state.

These distinctions emphasize the importance of tailoring public health strategies and policies to specific regional needs. Addressing disparities in healthcare accessibility, enhancing vaccination outreach in underserved communities, and improving chronic disease management are essential steps in mitigating the impact of ILI across different state types.

The feature importance (FI) scores obtained from the RFR model determine the relative weight ($w_k$), of each variable $k$ in computing the vulnerability index. The weight $w_k$ is a constant value applied uniformly to all states in the final index calculation. This weight value ($w_k$) combined with the normalized value ($x_{ik}$) of the indicator, gives the vulnerability index for each state ($V_i$) as given in Eq (3) in the methodology section. The top five variables, based on feature importance (FI) scores, included percentage of individuals moved from abroad, native and foreign-born populations without insurance, female population percentage, adult influenza vaccination rates, and the population over 65.

The observed relationships between the socio-economic indicators and ILI vulnerability are complex, with some indicators increasing vulnerability and others decreasing it (S2 Table). Indicators like moved from abroad [43], female [44,45], older (> 65) [46], Black/African American and Hispanic population (%) [47,48], and Population density [49], showed a positive correlation, which is consistent with established literature linking dense living conditions and systemic health disparities to faster disease transmission. Notably, all indicators related to the lack of health insurance (e.g., Native born No Insurance (%)) exhibited a negative correlation with the observed ILI rate. This counterintuitive finding may be attributed to testing or formal diagnosis for respiratory illnesses, leading to an artificially lower reported caseload in those states [50–52]. Similarly, indicators related to chronic conditions, such as 'Invasive cancer incidence', 'Asthma among adults (%)', and 'Diabetes among adults (%)', also showed a negative correlation with the ILI rate. Although populations with underlying long-term diseases are biologically more susceptible to severe outcomes, this negative association may result from heightened risk awareness and proactive behavior [53,54]. These high-risk individuals often adopt strict preventative measures and hygiene maintenance practices, thereby effectively reducing their overall chance of infection compared to the general population.

These findings highlight the crucial role of healthcare access, vaccination coverage, and socioeconomic disparities in influencing ILI vulnerability. Recent migration patterns and the presence of elderly populations also emerged as significant factors. This analysis underscores the necessity for targeted public health interventions that address insurance gaps, promote vaccination, and mitigate socioeconomic inequalities, while acknowledging the unique demographic and migration characteristics of each state. However, although this information provides us with an overall overview of the important factors to consider, a more localized, state-specific approach is essential. While the feature importance FI values of different factors provide valuable insight into their contribution to overall state vulnerability, it should not be interpreted as a universal standard for policymaking. The distribution of these parameters varies significantly across states, necessitating

PLOS Computational Biology

the development of tailored mitigation policies for each. Instead, the RFR-derived feature importance FI serves as a valuable tool for guiding future policy development, enabling targeted interventions that address state-specific vulnerabilities and disparities.

To understand the complex distribution of variable values across states from different vulnerability values, and to compare the variable values. Fig 4 illustrates the variation in socio-economic and health indicators across states, highlighting the complexity of state-level vulnerability. The size of each bubble proportionally reflects the normalized value of the indicator, ranging from 0 to 1, where larger bubbles indicate higher values. The color gradient of the bubbles denotes the

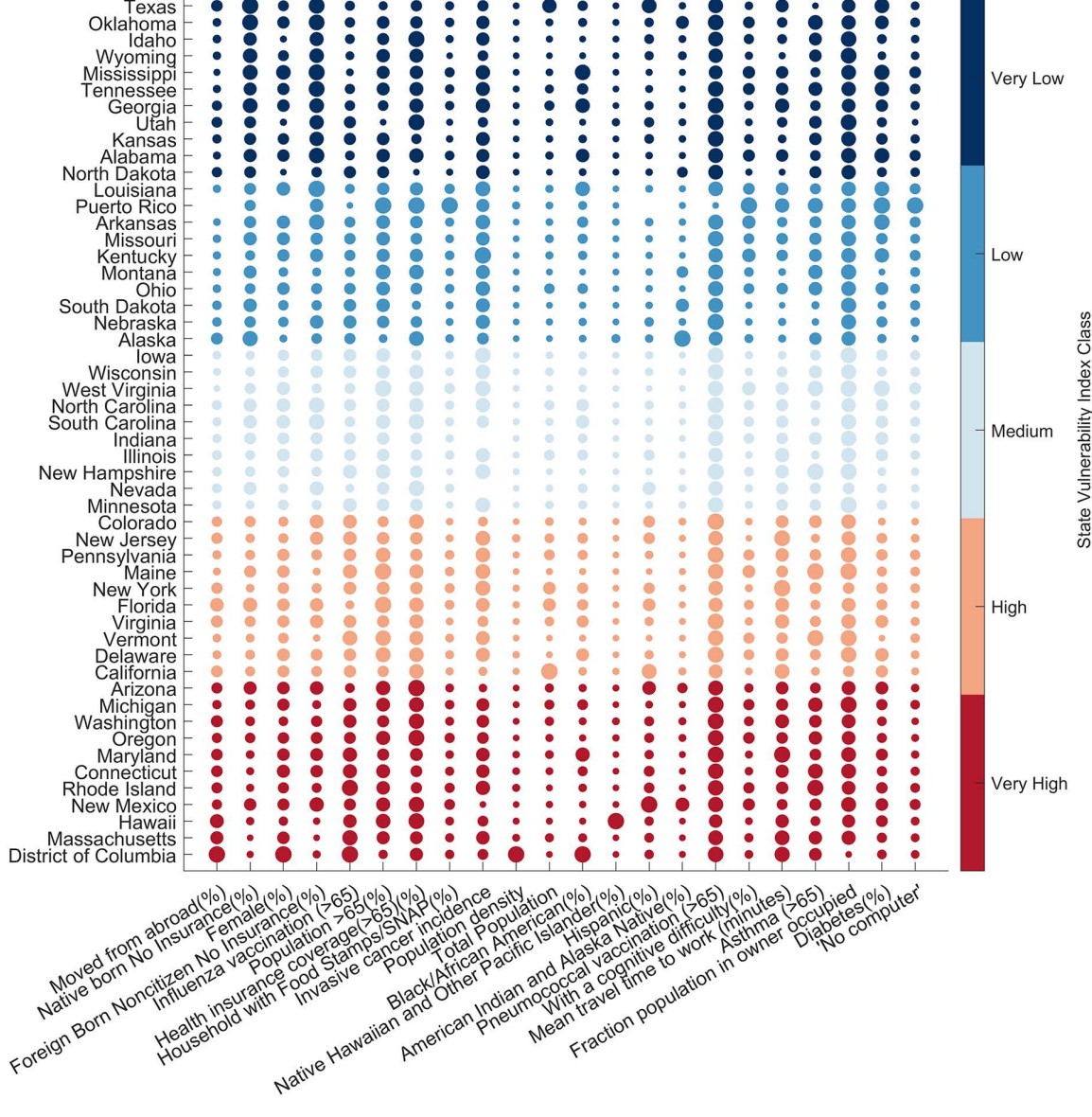

**Fig 4. State-level indicator trends and vulnerability.** This scatter plot displays normalized socio-economic and health indicators (0 to 1) across U.S. states. The size of each data point (bubble) corresponds to the normalized value of the indicator ($x_{ik}$), while the color represents the state's vulnerability class. The x-axis presents the indicators in descending order of RFR-derived feature importance or weight ($w_k$), ranging from "Moved from abroad (%)" (highest importance) to "No computer" (lowest importance).

normalized vulnerability index for each state, transitioning from dark red, representing high vulnerability, to blue, representing low vulnerability. High vulnerability in certain states does not necessarily result from higher values of all variables with greater weights, nor does low vulnerability imply lower values of these variables. Using the developed method, we identify specific factors contributing to a state's vulnerability.

Highly vulnerable states, such as Distict of Columbia (DC), Rhode Island, Maryland, and New Mexico, exhibit high vulnerability due to a combination of factors, including having been moved from abroad, female and aging populations, high rates of chronic illnesses (e.g., diabetes and cancer), and limited healthcare access. However, the contribution of specific indicators varies across these states. For example, population density has a high value (normalized value = 1) in D.C., but remains below 0.08 for most other highly vulnerable states. Similarly, the Black/African American population indicator shows high values for DC and Maryland, but remains below 0.32 for most other states. In contrast, the Hispanic population significantly influences vulnerability in New Mexico (normalized value = 1), whereas the normalized value remains below 0.33 in other highly vulnerable states. These variations underscore the need for state-specific policy interventions tailored to distinguish socio-economic and demographic contexts. The overall higher vulnerability is due to multiple factors with higher weights having a greater value.

Conversely, states with lower vulnerability demonstrate characteristics associated with higher people with insurance (normalized value > 0.5), lesser migration from abroad (normalized value < 0.4), and lower racial diversity (both African American and Hispanic). Texas shows normalized value 1 for Natives with no Insurance. People with insurance being a negative indicator results in lower vulnerability, as it is observed that people without insurance tend to avoid checkups and testing, causing lower reporting. These states tend to have lower population densities and total and percentage female populations. Notably, even in states characterized by lower overall vulnerability, ongoing management of specific variables is crucial to sustain resilience. While some states demonstrate lower overall vulnerability, often correlating with lower values in highly weighted variables, significant parameters can still be a matter of concern. For instance, several states (Texas, Oklahoma, Mississippi, Georgia, and Wyoming) exhibit high percentages (normalized value > 0.75) of both native and Foreign-Born Noncitizens with No Insurance, indicating a critical area for intervention. Furthermore, variability in the values of contributing factors is evident even among states classified within the same vulnerability category, as well as in other vulnerability categories. Although the categorization of states provides a useful framework for policy prioritization, a granular examination of individual parameters is imperative for developing targeted policy interventions and effective infection control strategies.

The analysis highlights the intricate interplay between socio-economic and health indicators in shaping state-level vulnerability to ILI. By leveraging the Random Forest Regression model, we not only identify the most influential factors driving vulnerability but also uncover the diverse ways in which these factors manifest across different states. The variability in indicator values across states with similar vulnerability scores underscores the necessity of tailored, data-driven policy interventions rather than one-size-fits-all solutions. This reinforces the importance of localized strategies that address state-specific risk factors. Ultimately, this study provides a framework for policymakers to better allocate resources, enhance public health preparedness, and implement targeted interventions that mitigate vulnerability and reduce health disparities across the U.S.

## 3. Discussion

Understanding regional variability is crucial for assessing the true impact of any disaster event, as different areas experience varying levels of exposure, vulnerability, and adaptive capacity. Vulnerability assessment, a widely used tool in hydroclimatic research of extreme weather events, helps quantify susceptibility to natural disasters and understand spatial heterogeneity in risk and impact. However, its application in health-related studies remains underexplored. In this study, we utilize the vulnerability framework to assess spatial variations in susceptibility to influenza-like Illness (ILI). By

integrating socio-economic and demographic factors, we aim to identify key drivers of elevated infection rates in hotspot areas, providing insights for targeted public health interventions.

The findings highlight the significant role of varying socio-economic conditions in shaping vulnerability to ILI across U.S. states. The study reveals that vulnerability is not uniform; rather, it is shaped by a combination of urbanization, demographics, healthcare access, and economic disparities. For example, the District of Columbia, characterized by high population density and mobility, faces increased risks due to a sizable uninsured foreign-born population and longer commute times. In contrast, rural states such as New Mexico, and Arizona exhibit heightened vulnerability due to aging, female and Hispanic populations. States like, Michigan, with its mix of urban and rural regions, face dual challenges—high transmission risks in cities and economic hardships in rural areas. Furthermore, the model assigned higher weights to population density, moved from abroad, female and older populations, and racial minority groups, highlighting their key contribution to increasing vulnerability. The importance of these factors underscores that preventative health measures are key determinants of state-level susceptibility. Conversely, the negative correlation found for the fraction of the population without insurance suggests that systemic barriers to healthcare access led to underreporting, thereby masking the true burden of disease in states with high uninsured populations. The ordered feature importance provides a data-driven roadmap for strategic public health and socio-economic policy intervention. Since the indicators are ranked by their influence on ILI vulnerability, government resources can be optimally directed toward the highest-leverage areas. Specifically, the strong importance of indicators related to health insurance coverage suggests that expanding access to affordable healthcare is necessary. High feature importance linked to the elderly, female, and African American populations necessitates specific, targeted public health programs to address the underlying social determinants that drive this age, gender, and race-specific risk, thereby working to close documented health disparity gaps. The results underscore the need for tailored public health measures, including expanded healthcare access in rural regions, targeted vaccination campaigns, improved insurance coverage, and initiatives addressing gender-specific healthcare disparities. By leveraging spatial vulnerability maps, policymakers can prioritize high-risk areas, allocate resources effectively, and develop proactive strategies to mitigate ILI risks. Stakeholders can implement effective measures to achieve the maximum reduction in state-level susceptibility to severe infectious disease outcomes by prioritizing the top-ranked structural and demographic deficiencies. Unlike traditional indices like SVI, which are generalized for all disasters, these vulnerability values provide more tailored information for ILI (S8 Fig).

This study demonstrates the variability of vulnerability to Influenza-like Illness (ILI) across different states. However, it is essential to acknowledge that this state-level aggregation is coarse and may not fully represent the actual socio-economic heterogeneity that exists within states. Socio-economic characteristics, and thus vulnerability, often vary significantly on a finer scale, such as the community or neighborhood level. While the resulting state-level vulnerability map is limited by the unavailability of finer-scale ILI data, the underlying feature-weighted index methodology remains a universal and generalizable framework for assessing relative susceptibility. Furthermore, the feature importance assigned to the key socio-economic indicators is hypothesized to reflect the fundamental mechanisms of vulnerability and can be extended to understand sub-state level variation even if the dependent variable data for that scale is currently absent. Also, the proposed socio-economic vulnerability quantification is inherently limited by the selection of indicators, since feature importance values are computed based on the socio-economic factors selected. From a methodological standpoint, the predictive performance of the Random Forest Regression (RFR) model is constrained by the small sample size (N = 50); incorporating a larger number of data points would be necessary to significantly improve its predictive accuracy.

The generated vulnerability maps serve as powerful visual tools, making it easier to communicate the spatial variability in susceptibility to ILI across different states. Beyond influenza, this methodology provides a scalable framework for assessing vulnerability to other infectious diseases, bridging traditional disaster risk assessment with epidemiological research. These vulnerability maps serve as critical decision-making tools for policymakers, enabling targeted resource allocation and the design of targeted public health policies. Insights from such studies can inform vaccination drives,

healthcare accessibility improvements, and socioeconomic support programs, ultimately strengthening pandemic preparedness and response efforts. Furthermore, this approach can be refined for finer-scale analyses, such as block-level or city-level vulnerability assessments, allowing for even more localized and effective public health strategies. While the current methodology successfully identifies static socio-economic drivers, future research should transition to a spatio-temporal modeling effort that incorporates the entire time series and multiple seasons to explicitly quantify the dynamic and temporal variability of ILI vulnerability. The RFR model effectively captures most of the spatially structured variability in ILI vulnerability, as validated by the significantly negative spatial autocorrelation. Nevertheless, future extensions of this work could benefit from exploring spatial or multilevel modeling frameworks, especially with finer-resolution data (e.g., county-level or census-tract level) and more variables (e.g., healthcare expenditure).

## 4. Methodology

To assess regional vulnerability to influenza-like Illness (ILI), this study employs a data-driven framework integrating socio-economic and demographic indicators. A systematic selection process is used to refine variables, ensuring they accurately capture factors that influence disease susceptibility. To address multicollinearity among predictors, the Variance Inflation Factor (VIF) is applied, retaining only independent variables for analysis. These indicators are normalized to a uniform scale and weighted based on their importance, derived from a Random Forest Regression model trained with k-fold cross-validation. This rigorous data-driven weighting, informed by the non-linear modelling capabilities of the Random Forest, aims to provide a more empirically grounded and potentially more accurate understanding of each indicator's contribution to regional vulnerability. The final vulnerability index is computed as a weighted sum of these normalized indicators, allowing for spatial comparisons and identification of high-risk regions (Fig 5). The details of the data curation and processing, and the details of the methodology are discussed in the following subsections.

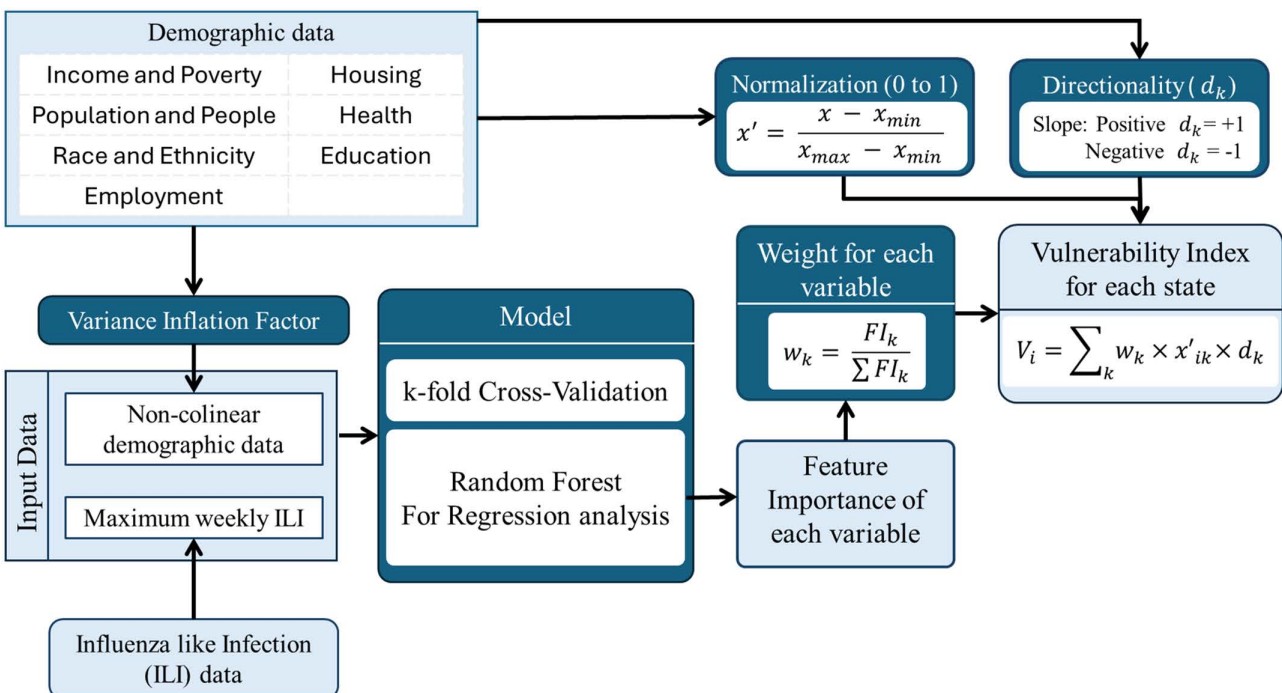

**Fig 5. Vulnerability Index Framework:** Schematic representation of the methodology for computing ILI vulnerability index ($V_i$) using a Random Forest regression method.

## 4.1. Data

Socio-economic vulnerability reflects a population's susceptibility to the impact of any crisis, including infectious disease outbreaks. Selecting appropriate indicators is crucial for assessing vulnerability of a region [16,55,56], as it ensures a data-driven evaluation of disease burden and healthcare disparities [57,58]. To quantify regional vulnerability to Influenza-like Illness (ILI), we incorporate diverse demographic and socio-economic factors that capture key aspects of societal resilience and risk. These indicators are sourced from publicly available datasets, from the U.S. Census Bureau (https://www.census.gov/) and the Centers for Disease Control and Prevention (CDC) (https://www.cdc.gov/fluview/), ensuring reliability and comparability. Given that socio-economic factors influence health outcomes through income disparities, healthcare access, housing conditions, and occupational risks, we categorized our selected indicators into seven key dimensions: (i)Income and Poverty, (ii)Population and People, (iii)Race and Ethnicity, (iv)Employment, (v) Housing, (vi)Health, and (vii)Education. Initially, over 450 variables were shortlisted based on their relevance and potential impact. However, with a sample size of only 50 states, retaining such a high number of variables posed a significant risk of overfitting. Therefore, we strategically reduced the variable count while preserving key information. This was achieved by considering corresponding broader variables instead of granular ones. For instance, total population and population over 65 are used instead of detailed age-specific population data, and state-level totals for income instead of age- and race-specific variables. This consolidation minimized redundancy and reduced model complexity. While finer-scale vulnerability analyses (e.g., county or ZIP code level) could benefit from more detailed variables, our state-level analysis necessitated a more controlled approach. Through a systematic selection process, based on past literature and data availability, we refined the list to 39 key indicators for all the states. The justification for these 39 variables, along with supporting references, is detailed in S1 Table, ensuring transparency and scientific rigor. To understand the relationship between these indicators as well as the dependent variable (percent ILI infection) we also examined the pair-wise correlation matrix (S2 Fig). Our examination reveals direct linear relationships between the independent and dependent variables, providing immediate interpretation for the final aggregate vulnerability index.

The number of Influenza-like Illness (ILI) cases for each state is obtained from the CDC FluView data, which provides weekly infection counts at the state level. To assess the maximum burden on the healthcare system, we analyze peak infection levels, as they represent the highest demand for medical resources. The maximum weekly infection observed throughout the year is considered to quantify the vulnerability of each state. Since absolute case numbers can be misleading due to population differences, we normalized the data by expressing the maximum weekly infection count as a percentage of the total state population, allowing for a standardized comparison of disease burden across states. This metric, defined as the maximum weekly infections (in percentage) for each week in 2022 (Fig 1), provides insight into the worst-case scenario for healthcare strain, facilitating a better understanding of regional vulnerabilities and preparedness.

## 4.2. Data preprocessing

Demographic and health variables often exhibit high collinearity, which can compromise the reliability and interpretability of machine learning models by leading to unstable coefficients and overfitting. To address this, we applied the Variance Inflation Factor (VIF), a statistical measure that quantifies the extent to which collinearity inflates the variance of regression coefficients. VIF is calculated as $1/(1 - R^2)$, where $R^2$ represents the coefficient of determination when a predictor is regressed against all other predictors.

While Random Forest Regression (RFR) models are inherently robust to multicollinearity and do not strictly require pre-processing, such as the Variance Inflation Factor (VIF) for prediction, the presence of highly correlated predictor variables can still destabilize the interpretation of feature importance [59,60]. Given that socio-economic vulnerability indicators often exhibit strong internal collinearity, we aimed to select a feature set that offers a clearer, and more independent understanding of the contribution to Influenza-like Illness (ILI) vulnerability. To filter out highly collinear variables and ensure the statistical independence of our final predictor set, we incorporated VIF as a pre-processing step. We analyzed

the VIF values for all 39 indicators in S3 Fig. To ensure statistical independence of the final predictor set, a VIF threshold of 10 was employed, a widely accepted practice in statistical modeling for identifying high multicollinearity [61,62]. After systematically removing variables exceeding this threshold, the resulting feature set was reduced from 39 to 22 variables. The VIF values for these remaining non-collinear variables are presented in S4 Fig.

The variables used in the analysis have different units and scales, which can introduce biases and disproportionate influences in the final vulnerability index. Therefore, all the input variables are normalized, allowing each factor to contribute proportionally without being dominated by larger numerical values. This normalization removes the effect of units and dimensionality, ensuring that no single variable skews the results. All selected demographic indicators ($x$) are transformed to a standardized $0 - 1$ scale using min-max normalization, as shown in Equation (1):

$$x' = \frac{x - x_{min}}{x_{max} - x_{min}}$$

(1)

Where, $x'$ = normalized variable $x$

$x_{max}$ = maximum value of variable $x$ across all states

$x_{min}$ = minimum value of variable $x$ across all states

## 4.3. Model development

A Random Forest Regression (RFR) model is applied with k-fold cross-validation to explore the relationship between demographic factors and ILI cases. The use of cross-validation minimizes overfitting and ensures robustness in the model's predictions. RFR is particularly well-suited for this analysis as it accommodates non-linear relationships and complex interactions among socio-economic indicators, with epidemiological data. Additionally, its ensemble nature enhances predictive accuracy and reduces sensitivity to noise, making it a reliable tool for assessing vulnerability. The feature importance (FI) of each variable is then extracted from the trained Random Forest model, quantifying the relative contribution of each predictor in explaining variations in ILI cases.

To optimize model performance, we experimented with different numbers of decision trees, ranging from 25 to 500, and evaluated their impact on prediction accuracy. The optimal number of Random Forest decision trees was determined using repeated 10-fold cross-validation across 200 simulation runs. The optimal number of trees was found to be 450, based on the stabilization of the averaged out-of-sample metrics: RMSE = 0.3886 and MAE = 0.3174. The stability analysis for the number of trees, ranging from 25 to 500 (in increments of 25), is detailed in S5 and S6 Figs. The 22 variables chosen based on the lowest Variance Inflation Factor (VIF) values were then used as input features.

The Random Forest Regression (RFR) model is employed to determine the relative contribution of each socio-economic indicator. To provide a robust, out-of-sample estimate of predictive performance and to mitigate bias from a single split, we employed a repeated random holdout validation. This involved running 100 iterations, each utilizing an 80%/20% split of the data for training and testing, respectively. The resulting average out-of-sample performance has a Root Mean Squared Error (RMSE) of 0.418. However, the primary objective remains the stable identification and weighting of key vulnerability factors. The model successfully achieved this by yielding stable, non-zero feature importance scores for the final set of non-collinear predictors.

The RFR model calculates feature importances (FIs), which quantify the relative contribution of each variable to the model's predictive accuracy. The FI values corresponding to each indicator are presented in S7 Fig. This score does not directly indicate the vulnerability value. The FI values are used to compute weights ($w_k$) assigned to each vulnerability indicator ($k$) to quantify their impact on the final vulnerability index. These weight values remain constant for all the states. The weight for each variable ($w_k$) is computed using Equation (2):

$$w_k = \frac{FI_k}{\sum FI_k}$$

(2)

Where, $w_k$ = Weight assigned to each vulnerability indicator ($k$)

  $FI_k$ = Feature Importance of vulnerability indicator ($k$)

### 4.4. Directionality of vulnerability indicators

While the Random Forest Regression (RFR) model accurately quantifies the relative importance ($FI_k$) of each socio-economic indicator, this metric alone does not reveal the nature of the relationship (i.e., whether an increase in the indicator value increases or decreases vulnerability). To establish the functional relationship between each factor and the predicted ILI rate, we employed the Partial Dependence Plots (PDPs) technique. Based on the slope of the PDP curve, we assigned a Directionality Factor ($d_k$) to each indicator:

  A value of $d_k = +1$ was assigned if the indicator has a positive relationship and is increasing vulnerability, and $d_k = -1$ if the indicator has a negative relationship and is decreasing vulnerability. The $d_k$ values corresponding to each indicator are provided in S2 Table.

### 4.5. Vulnerability calculation

The final Vulnerability Index ($V_i$) for each state is computed using a weighted summation of the selected, normalized variables, and directionality as expressed in Equation (3). This approach ensures that the vulnerability score not only reflects the magnitude of each contributing factor but also incorporates its relative importance and directionality in influencing the spatial variability of ILI. Vulnerability at the state level was determined by the linear combination of weighted variable values, where the weights were derived from the FI scores.

$$V_i = \sum_k w_k \times x'_{ik} \times d_k \tag{3}$$

Where, $w_k$ = Weight assigned to each vulnerability indicator ($k$)

  $V_i$ = Vulnerability Index at each state ($i$)

  $x'_{ik}$ = Value of variable $k$ for state $i$

  $d_k$ = Directionality of variable $k$

  $d_k$ = 1 for positive slope

  - 1 for negative slope

## Supporting information

**S1 Table. List of variables selected as indicators of the socio-economic vulnerability with respect to ILI along with justification of selection of each variable.**
(DOCX)

**S2 Table. Socioeconomic indicators and directions computed using Partial Dependence Plots.**
(DOCX)

**S1 Fig. LISA (Local Moran's I) scatter plot of RFR model residuals across 52 U.S. states.** The negative slope (Moran's I = −0.0435, p = 0.0041) indicates significant spatial dispersion, suggesting that neighboring states tend to exhibit dissimilar residual values, with no evidence of spatial clustering.
(PNG)

**S2 Fig. Pairwise Correlation Matrix for all 23 variables considered (22 independent input variables and 1 dependent variable (percent ILI infection)) with color intensity indicating the strength and direction of the relationship.**
(TIF)

**S3 Fig. VIF values for all 39 indicators.**
(TIF)

**S4 Fig. VIF values for the final 22 indicators.** All the variables have values less than 10.
(TIF)

**S5 Fig. The root means square (RMSE) value corresponding to different number of decision trees selected for the RFR model.**
(TIF)

**S6 Fig. The mean absolute error (MAE) value corresponding to different number of decision trees selected for the RFR model.**
(TIF)

**S7 Fig. Final Set of Predictor Variables and their feature Importance for Influenza Vulnerability computed using random forest.**
(TIF)

**S8 Fig. Comparison of Vulnerability and CDC SVI Classes by State.**
(TIF)

## Acknowledgments

The authors thank Dr. Joshin Kumar from Washington University for insightful discussions.

## Author contributions

**Conceptualization:** Shrabani S. Tripathy, Rajan K. Chakrabarty.

**Data curation:** Shrabani S. Tripathy, Taveen S. Kapoor.

**Formal analysis:** Shrabani S. Tripathy.

**Funding acquisition:** Joseph V. Puthussery, John R. Cirrito, Rajan K. Chakrabarty.

**Methodology:** Shrabani S. Tripathy.

**Project administration:** Joseph V. Puthussery.

**Supervision:** John R. Cirrito, Rajan K. Chakrabarty.

**Visualization:** Shrabani S. Tripathy, Taveen S. Kapoor.

**Writing – original draft:** Shrabani S. Tripathy.

**Writing – review & editing:** Joseph V. Puthussery, Taveen S. Kapoor, John R. Cirrito, Rajan K. Chakrabarty.

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
