## [Decision Letter · Decision Letter 0]

7 Sep 2025

Spatial variation in socio-economic vulnerability to Influenza like infection for the US population

PLOS Computational Biology

Dear Dr. Chakrabarty,

Thank you for submitting your manuscript to PLOS Computational Biology. After careful consideration, we feel that it has merit but does not fully meet PLOS Computational Biology's publication criteria as it currently stands. Therefore, we invite you to submit a revised version of the manuscript that addresses the points raised during the review process.

Please submit your revised manuscript within 60 days Nov 07 2025 11:59PM. If you will need more time than this to complete your revisions, please reply to this message or contact the journal office at ploscompbiol@plos.org. Please include the following items when submitting your revised manuscript:

We look forward to receiving your revised manuscript.

Kind regards,

Samuel V. Scarpino

Academic Editor

PLOS Computational Biology

Denise Kühnert

Section Editor

PLOS Computational Biology

**Additional Editor Comments:**

I agree with the reviewers that this is an interesting study that is likely of wide interest. As the authors note, we typically struggle to integrate measures of social vulnerability because they vary significantly over finer geographic scales than those associated with most of our infectious disease data. The reviewers provide a number of thoughtful comments that I strongly suggest the authors pay careful attention to during their revision. Looking across the reviewer comments, and considering my own evaluation, there is consensus that more work demonstrating the applicability of the aggregate social vulnerability measures would strengthen the manuscript considerably. I would encourage the authors to consider directly comparing their measure to the CDC's Social Vulnerability Index (accomplishing this may require aggregating that measure appropriately). I agree with R2's comments that showing some pairwise relationships between components of the new aggregate measure and disease would aid with interpretation. More importantly, I am interested in how sensitive the results are to level of aggregation. While I realize you cannot look at ILI data sub-state-level, you can do so with the social vulnerability data. What would the results look like if you ran a kind of bootstrap strap analysis where you randomly selected census tracts to create 50 "psuedo-states" and use the resulting random forest models as a kind of null distribution? Could you train the model on 80% of the census tracts in states and predict aspects of social vulnerability at the census tract level in the 20% of census tracts that were held out? In terms of variable reduction, I would recommend comparing your method to commonly used dimensionality reduction approaches like principal component analysis. To this last point, it's not obvious to me why you need to perform dimension reduction when using a regularized approach like random forest (I can think of plausible reasons, but it would be useful to explain directly). I did see that the authors referenced some supplemental material, but I was unable to access any supplement. Lastly, the authors state that data is publicly available. While it's true that data are publicly available, the intent behind the journal's policies are to facilitate reproducibility and allow others to effectively build from your work. I would strongly encourage the authors to make their data and analysis scripts publicly available via Github and archived with a DOI for some service like figshare or zenodo.

**Journal Requirements:**

1) Please upload all main figures as separate Figure files in .tif or .eps format. For more information about how to convert and format your figure files please see our guidelines:

2) We have noticed that you have uploaded Supporting Information files, but you have not included a list of legends. Please add a full list of legends for your Supporting Information files after the references list.

3) Some material included in your submission may be copyrighted. According to PLOSu2019s copyright policy, authors who use figures or other material (e.g., graphics, clipart, maps) from another author or copyright holder must demonstrate or obtain permission to publish this material under the Creative Commons Attribution 4.0 International (CC BY 4.0) License used by PLOS journals. Please closely review the details of PLOSu2019s copyright requirements here: PLOS Licenses and Copyright. If you need to request permissions from a copyright holder, you may use PLOS's Copyright Content Permission form.

Potential Copyright Issues:

i) Figures 1, and 2. Please (a) provide a direct link to the base layer of the map (i.e., the country or region border shape) and ensure this is also included in the figure legend; and (b) provide a link to the terms of use / license information for the base layer image or shapefile. We cannot publish proprietary or copyrighted maps (e.g. Google Maps, Mapquest) and the terms of use for your map base layer must be compatible with our CC BY 4.0 license.

4) Please amend your detailed Financial Disclosure statement. This is published with the article. It must therefore be completed in full sentences and contain the exact wording you wish to be published.

5) Please send a completed 'Competing Interests' statement, including any COIs declared by your co-authors. If you have no competing interests to declare, please state "The authors have declared that no competing interests exist". Otherwise please declare all competing interests beginning with the statement "I have read the journal's policy and the authors of this manuscript have the following competing interests:"

**Reviewers' comments:**

Reviewer's Responses to Questions

**Comments to the Authors:**

Reviewer #1: The manuscript presents a novel machine-learning-driven framework to assess and map state-level socio-economic vulnerability to Influenza-Like Illness (ILI) in the United States. By integrating 39 socio-economic and health indicators from U.S. Census and CDC data, the authors develop a vulnerability index using Random Forest Regression (RFR) to quantify regional disparities in ILI susceptibility. The study’s focus on socio-economic determinants of health (SDOH) and its application of a data-driven approach to public health vulnerability assessment is a significant contribution to the field. The use of spatial mapping to visualize vulnerability enhances the practical utility of the findings for policymakers. However, several areas require clarification or improvement to strengthen the manuscript’s methodological rigor, interpretative caution, and broader applicability.

The use of Random Forest Regression to weight socio-economic and health indicators for a vulnerability index is a robust and replicable approach. The methodology leverages the non-linear modeling capabilities of RFR, which is well-suited for capturing complex interactions among SDOH. The inclusion of 39 diverse indicators across seven dimensions (Income and Poverty, Population and People, Race and Ethnicity, Employment, Housing, Health, and Education) provides a holistic view of vulnerability, grounded in established SDOH literature.

The manuscript occasionally employs language that implies causal relationships. Random Forest models assess associations, not causation, and such statements risk overinterpretation. The authors should revise these claims to emphasize correlations and clarify the limitations of RFR in establishing causality. While the manuscript mentions the use of K-fold cross-validation and the selection of 200 decision trees based on minimized RMSE (Page 21, Line 373), it does not report specific model performance metrics (e.g., RMSE, R², or Mean Absolute Error) for the final RFR model. Including a dedicated model validation section with quantitative metrics (e.g., test-set predictions, cross-validation scores, or bootstrapping results) would strengthen confidence in the model’s predictive accuracy and generalizability.

The rationale for selecting the initial 39 indicators and reducing them to 22 using Variance Inflation Factor (VIF) thresholding is insufficiently detailed (Page 20, Lines 319–349). The manuscript does not discuss the implications of excluding specific variables or justify the VIF threshold of 10. A more transparent explanation, supported by references or sensitivity analyses, would enhance the credibility of the variable selection process. Additionally, Supplementary Table 1, referenced for justification, is not provided in the document, limiting the ability to evaluate the scientific rigor of the selection.

The study’s geographic framing would benefit from testing for spatial autocorrelation in model residuals, using metrics such as Moran’s I or Local Indicators of Spatial Association (LISA). This analysis would confirm whether the RFR model fully captures regional variations in ILI vulnerability or if unmeasured spatial processes (e.g., regional policy differences, healthcare infrastructure gaps, or cultural factors) influence the results. Significant spatial clustering in residuals could suggest the need for spatial modeling (e.g., Geographically Weighted Regression) or multilevel approaches in future work.

Aggregating SDOH at the state level may obscure significant intra-state variations, particularly in large or socio-economically diverse states like California, Texas, or New York (Page 9, Line 123). While the authors acknowledge the potential for finer-scale analyses (e.g., county or ZIP code level), they do not discuss how state-level aggregation might mask local disparities, which is a critical limitation in SDOH research. For example, socio-economic factors like poverty or healthcare access often operate at the community or county level, and their impact may be diluted or misrepresented at the state level. The manuscript should include a discussion of this limitation and consider evaluating the relevance of certain indicators (e.g., population density, healthcare access) at the state level versus finer scales.

The manuscript makes a compelling contribution to public health by introducing a machine-learning-driven framework for assessing socio-economic vulnerability to ILI. The use of RFR and spatial mapping is innovative and has significant potential for informing targeted interventions. However, addressing the concerns regarding causal language, model validation, indicator selection, spatial autocorrelation, and state-level aggregation will significantly strengthen the manuscript’s scientific rigor and impact. With these revisions, the study could serve as a valuable resource for public health researchers and policymakers aiming to mitigate infectious disease disparities.

Reviewer #2: Overall, the authors attempt to answer an important questions through a novel framework. My biggest concern is around the assumption of direction, more or less vulnerable, based on variable importance outputted from random forest regression. Random forests do not allow for interpretation of relationship, i.e. although a variable is important to the outcome it is not clear if it has a positive or negative relationship. My suggestion would be to add more descriptions or supplemental tables/images showing the relationship between each indicator used to create the social vulnerability index and the main outcome of the regression model, I think ILI, by doing univariate analysis or even multivariate since multicollinearity is taken care of in variable selection. This will assure the reader that the assumed “more vulnerable” relationship is valid.

Abstract

-“To address multicollinearity, VIF was applied.” Do you mean “to assess” and “was calculated”?

-I suggest adding the timeframe of the study into the abstract, what influenza season?

Introduction

-I suggest adding an example to socio-economic structures on line 56 in the Introduction. A simple “,like …..” may be nice for the readers.

-I suggest defining more clearly what “regions” your study looks at in the introduction’s 3rd paragraph, i.e. state. Region is quite vague to be the only term used in that paragraph.

-I suggest moving the 5th paragraph in the introduction to be added on the 2nd paragraph since it nicely gives the introduction on ILI and vulnerability all at once.

-I suggest adding your timeframe into the last paragraph of the introduction, i.e. what time frame your study looks at in the US.

-I recommend either discussing the make up of ATSDR’s social vulnerability index in the introduction or in the discussion for comparison. https://www.atsdr.cdc.gov/place-health/php/svi/index.html

Results

-I suggest adding what the regression model’s main outcome was. I think it helps clarify for the reader to understand the weights.

-It seems that figure 1 goes with the method description, which is after the results section. Please align images with current text placement.

-Figure 3’s legend needs more description about what the numbers are. Weights or proportions or feature importance? It is unclear.

-On lines 209-211, you refer to figure 4 but also to weights. Are there weights in figure 4? It is unclear if the normalized value of the indicator is the same as weights or different or what that relationship is.

-I suggest presenting a visual, table or supplement to explain the importance and weights for each indicator.

Discussion

-Please add limitations.

-Since you did not have other respiratory indicators in the random forest, like weather or schooling data or mobility data, then you cannot say that this probes socio-economic indicators shape ILI more than any other indicators. This is because you only looked at socio-economic indicators. Therefore, you can only have relative importance noted between the indicators you investigated. For example, if you had put in humidity data and that was the most important variable then the message would be that humidity is more important than socio-economic. Just make sure all of the discussion and results are relative to only the indicators you investigated. It may be worth it to note this in the limitations.

-Limitation to interpret the variables relationship to ILI from a random forest regression. For example, % female may have a string relationship with ILI in a positive way, i.e. the lower the proportion the higher the ILI. It should be clear that the limitation from random forest variable importance is not being able to infer a relationship direction.

Methodology

-Be very careful with interpreting variable importance as indicating vulnerability.

-Do you use 1 peak week data for each state? So the data going into the model is only 50 observations? Why not use the entire time series? Please explain in the methods. Also, I would suggest using multiple seasons then.

Reviewer #3: General Comments

The present study aims to quantify environmental health impacts and assess risk by examining the disproportionate burden of Influenza-Like Illness (ILI). While the manuscript has potential and offers novel contributions, it requires substantial revisions before it can be considered for publication.

Major Comments

The introduction is too general. It would be more effective if the authors focused on the burden of flu-like diseases. The magnitude and significance of the problem should also be clearly articulated in the introduction.

The manuscript does not adequately address the rich body of existing literature on SVI and influenza interventions (e.g., vaccination). To strengthen the paper, key references should be incorporated. For instance, the following works highlight the magnitude and significance of the problem and may be added:

Tatar, M., Faraji, M. R., & Wilson, F. A. (2023). Social vulnerability and initial COVID-19 community spread in the US South: A machine learning approach. BMJ Health & Care Informatics, 30(1), e100703.

O’Sullivan, T., & Bourgoin, M. (2010). Vulnerability in an influenza pandemic: Looking beyond medical risk. Behaviour, 11(16).

Nayak, A., Islam, S. J., Mehta, A., Ko, Y. A., Patel, S. A., Goyal, A., ... & Quyyumi, A. A. (2020). Impact of social vulnerability on COVID-19 incidence and outcomes in the United States. MedRxiv, 2020-04.

Khazanchi, R., Beiter, E. R., Gondi, S., Beckman, A. L., Bilinski, A., & Ganguli, I. (2020). County-level association of social vulnerability with COVID-19 cases and deaths in the USA. Journal of General Internal Medicine, 35(9), 2784–2787.

While the authors acknowledge the economic burden of influenza, the model itself does not properly include crucial economic variables such as healthcare expenditure.

Given the presence of a powerful and well-defined index such as the SVI, the authors need to better justify their choice of variables. Although an attempt was made to explain the rationale for variable selection, some choices may not be defensible. For example, “No computer” may not be a strong indicator, as most individuals in the U.S. have access to cell phones, which arguably provide even greater access to information for the general population.

The discussion section lacks an in-depth analysis of how socioeconomic factors influence the results.

**Have the authors made all data and (if applicable) computational code underlying the findings in their manuscript fully available?**

Reviewer #1: Yes

Reviewer #2: Yes

Reviewer #3: None

PLOS authors have the option to publish the peer review history of their article (what does this mean? ). If published, this will include your full peer review and any attached files.

**Do you want your identity to be public for this peer review?** For information about this choice, including consent withdrawal, please see our Privacy Policy .

Reviewer #1: No

Reviewer #2: No

Reviewer #3: No

**Figure resubmission:**
---

## [Decision Letter · Decision Letter 1]

12 Dec 2025

Dear Chakrabarty,

We are pleased to inform you that your manuscript 'Spatial variation in socio-economic vulnerability to Influenza like infection for the US population' has been provisionally accepted for publication in PLOS Computational Biology.

Best regards,

Samuel V. Scarpino

Academic Editor

PLOS Computational Biology

Denise Kühnert

Section Editor

PLOS Computational Biology

Reviewer's Responses to Questions

**Comments to the Authors:**

Reviewer #2: Thank you for addressing previous comments so thouroughly with additional analysis. I believe the paper is ready to be published.

Reviewer #3: Na

**Have the authors made all data and (if applicable) computational code underlying the findings in their manuscript fully available?**

Reviewer #2: None

Reviewer #3: Yes

PLOS authors have the option to publish the peer review history of their article (what does this mean? ). If published, this will include your full peer review and any attached files.

**Do you want your identity to be public for this peer review?** For information about this choice, including consent withdrawal, please see our Privacy Policy .

Reviewer #2: No

Reviewer #3: No

---

## [Editor Report · Acceptance letter]

PCOMPBIOL-D-25-01042R1

Spatial variation in socio-economic vulnerability to Influenza-like infection for the US population

Dear Dr Chakrabarty,

I am pleased to inform you that your manuscript has been formally accepted for publication in PLOS Computational Biology. Your manuscript is now with our production department and you will be notified of the publication date in due course.

With kind regards,

Judit Kozma
